# An Analysis of the Social and Economic Costs of Breast Cancer in Italy

**DOI:** 10.3390/ijerph18179005

**Published:** 2021-08-26

**Authors:** Francesco Saverio Mennini, Marco Trabucco Aurilio, Simone Gazzillo, Claudia Nardone, Paolo Sciattella, Andrea Marcellusi, Raffaele Migliorini, Valerio Sciannamea, Andrea Piccioni, Matteo Bolcato, Sandro Barni

**Affiliations:** 1EEHTA-CEIS, DEF Department, Faculty of Economics, University of Rome “Tor Vergata”, 00133 Rome, Italy; mennini@uniroma2.it (F.S.M.); simone.gazzillo@uniroma2.it (S.G.); claudia.nardone@uniroma2.it (C.N.); paolo.sciattella@uniroma2.it (P.S.); mrcndr00@uniroma2.it (A.M.); 2Institute for Leadership and Management in Health, Kingston University, London KT1 2EE, UK; 3Department of Medicine and Health Sciences “V. Tiberio”, University of Molise, 86100 Campobasso, Italy; marco.trabuccoaurilio@unimol.it; 4Office of Medical Forensic Coordination, Italian National Social Security Institute (INPS), 0144 Rome, Italy; raffaele.migliorini@inps.it (R.M.); valerio.sciannamea@inps.it (V.S.); 5Emergency Medicine, Gemelli, IRCCS (Scientific Institute for Hospitalization and Treatment), 00168 Rome, Italy; andrea.piccioni@policlinicogemelli.it; 6Legal Medicine, University of Padua, Via G. Falloppio 50, 35121 Padua, Italy; 7Local Health Unit—ASST Bergamo Ovest, 24047 Treviglio, Italy; sandro.barni@ospedale.treviglio.bg.it

**Keywords:** breast cancer, health economics, social security, hospital

## Abstract

Background: Breast cancer is the most prevalent cancer affecting women and it represents an important economic burden. The aim of this study was to estimate the socio-economic burden of breast cancer (BC) in Italy both from the National Health Service (NHS) and the government perspectives (costs borne by the social security system). Methods: The economic analysis was based on the costs incurred by the NHS from 2008 to 2016 (direct costs related to hospitalizations) and by the National Social Security Institute (INPS) from 2009 to 2015 (costs of social security benefits) for patients with breast cancer. The analysis was based on the Hospital Information System (HIS) and Disability Insurance Awards databases. For both databases, patients affected by a malignant neoplasm of the female breast, carcinoma in situ, or secondary malignant neoplasm of the breast were considered. Results: Results show that more than 75,000 women were hospitalized for breast cancer every year, with an overall cost for hospitalization of about €300 million per year. From the Social Security analysis, a number of 29,000 beneficiaries each year was estimated. Considering per patient social costs, breast cancer at the primary stage cost €8828 per year, while secondary neoplasms cost €9780, with an average total economic burden of €257 million per year. Conclusions: This analysis focused on the economic impact of breast cancer in Italy, showing that an advanced stage of the disease was associated with a higher cost.

## 1. Introduction

Breast cancer (BC) is the most common cancer in women worldwide (and the second most common cancer overall, with nearly 2.1 million new cases diagnosed in 2018. This represents about 12% of all new cancer cases and 25% of all cancers in women [1,2,3,4]. BC is the most diagnosed neoplasm in women in Italy, after skin cancer. It covers about 30% of all female malignant cancers. Today, in Italy, there are about 800,000 women who have been diagnosed with breast cancer, equating to 44% of all women with a previous diagnosis of cancer, and it represents a major public health problem from the NHS perspective [1]. Despite the increasing prevalence of the disease, mortality is decreasing in all age groups, particularly in women under 50. In Italy, the survival rate of women with BC is 87% within 5 years after diagnosis and 80% within 10 years. However, in 2016, breast cancer was the leading cause of cancer-related deaths in women, with more than 12,000 deceased. The impact of this disease is clear not only in terms of mortality and morbidity but also in terms of economic consequences for all National Health Services (NHSs) and from a social point of view [1,2,3,4].

According to a 2018 Dutch study, in the Netherlands BC causes 3100 deaths, 26,000 years of life lost, and 65,000 DALYs every year, with a total expense of about €1.27 billion. In addition, in the Netherlands from 1990 to 2014, there was an increase in the incidence of BC, from 103.4 to 153.2 per 100,000 women [5]. This rise is apparently due to the increased incidence of ductal in situ carcinoma (stage 0) and early breast cancer (stage I). The incidence of advanced BC (stages II–III) and metastatic BC (stage IV) was steady in the observation period. The study also showed an increase in 10-year survival rates in the Netherlands [5].

On the other hand, in the United States, a review on 29 Cost of Illness assessments conducted by Campbell was published in 2009 to quantify the cost of BC treatment. Every patient suffering from breast cancer in their lifetime involves a cost ranging from $20,000 to $100,000 [6]. As far as surgeries are concerned, they had a similar expense in the case of both mastectomy and conservative surgery. The authors estimate that there was a significant incremental rate, from $23,000 to $31,000, for patients who were treated with adjuvant chemotherapy [6].

This link between the cost of treatment and the stage of the disease is also confirmed by a systematic review of the literature published in 2018 and based on 20 studies (15 from high-income countries and 5 from low/middle-income countries). The comparison shows that the average treatment cost of BC in stages II, III, and IV is 32%, 95%, and 109%, respectively, higher than the treatment cost of BC in stage I [7].

In 2019, Piccinni published a study on BC starting from a real-world administrative healthcare database. The study considered patients with hormone breast cancer (HR+/HER2-), analyzing a follow-up period of two years to quantify the health costs for the NHS. These costs amounted, on average, to €7543 in the first year and €4834 in the second year of follow-up [8].

Moreover, especially in a public system such as Italy, an important economic burden was registered also in terms of disability benefits. In Italy, a study by the National Institute of Statistics (ISTAT) [9] estimated almost 1.5 million beneficiaries of social security benefits totally in 2013, with a cost of €15.7 billion. On the other hand, in 2015, the National Social Security Institute reported 1.1 million private or self-employed workers receiving social security benefits totally, and 17 million workers insured against the risk of becoming unable to work due to a physical or mental disability [10].

A study conducted by the Legal-Medical Coordination Office of the National Social Security Institute estimated [11] that in 2012 the group of diseases denominated as “Tumours” was the first in the number of incapacity pensions approved, amounting to over 130,000 (28% of all pensions recognized). In particular, 18,627 approved applications were registered for breast cancer [12]. 

The studies mentioned were reported to explain the impact that such a widespread pathology as breast cancer has in different countries in terms of costs, epidemiology, and disease progression.

The Italian health care system is a mixed public–private system. The National Health Service guarantees universal and free of charge coverage for all citizens and legal foreign residents. It is funded by corporate and value-added tax revenues collected by the central government and distributed to the regional governments, which are responsible for delivering care. Residents receive mostly free primary care, inpatient care, and health screenings [13].

Since the NHS does not allow people to opt out of the system and seek only private care, substitutive insurance does not exist, and complementary and supplementary private health insurance play a limited role in the health system.

On the other hand, the social security system provides, upon request, economic benefits to all workers whose working capacity is reduced or absent due to physical or mental illness, largely financed by their contributions.

The objective of this study was to estimate the socio-economic burden of breast cancer in Italy, considering both the NHS and social security system perspectives. For the first one, direct costs related to hospitalizations were considered, through a real-world data analysis, based on data from the Hospital Information System (HIS). Together with healthcare direct costs, social security costs are an important component and must be considered in economic evaluations to define the total burden of the disease. This analysis was based on the Disability Insurance awards database and estimated the number of disability benefit receipts suffering from breast cancer and related costs.

Then, the analysis was focused on separately considering different stages of the disease: primary/in situ BCs, which concern an initial stage of the disease; primary BC with progression; and secondary breast cancer, occurring at an advanced stage of the disease. Therefore, the objective was to analyze the cost difference on the basis of the presence or absence of early diagnosis and early care of the patient.

## 2. Materials and Methods

### 2.1. Estimation of Hospitalization Costs

The analysis of hospital care for the treatment of breast cancer was conducted using data from the Hospital Information System (HIS), and in particular, the hospital discharge records (HDR), available for the period between 2008–2016. These records provided information on all hospital admissions—ordinary and day-hospital (DH) ones—throughout the country. Day-hospital admissions refer to those without an overnight stay. In particular, the hospital activity for each year was described, in terms of hospitalizations, expenditures incurred by the NHS, and prevalent patients.

The patients suffering from breast cancer during the study period were selected based on considering all the subjects with at least one hospital admission with a primary or secondary diagnosis of primary carcinoma (diagnosis code ICD 9 CM 174.xx), in situ carcinoma (diagnosis code ICD 9 CM 233.0), or secondary carcinoma (diagnosis code ICD 9 CM 198.81). The first two diagnoses can be associated with an early stage of the disease, while secondary carcinoma refers to an advanced stage.

The impact on the NHS of hospital care related to the treatment of the disease was estimated by considering all admissions due to breast cancer, chemotherapy (Diagnosis Related Groups - DRG 410), or radiotherapy (Diagnosis Related Groups - DRG 409), per single year. The theoretical valuation of the hospitalizations was estimated on the basis of the assumption that each hospitalization is remunerated according to the values of the national rates (Ministerial Decree 12 September 2006 and Ministerial Decree 18 October 2012). The DRG system aggregates all activities, including surgeries, drugs administered, materials, and personnel paid out on each individual diagnosis and defines the reimbursement rate. This value corresponds to the total amount of resource utilization, which must be reimbursed to the hospital. Under the DRG-based reimbursement system, each hospitalized patient is allocated to a group of homogeneous diagnostic cases. Therefore, patients with the same DRG value have been allocated the same reimbursement costs, which represents an average value of resource utilization between hospitalizations attributable to that DRG.

Finally, a specific analysis was performed for the HIS database for a 5-year follow-up period, stratifying the population into “primary with no progression”, “primary with progression”, and “secondary BC” patients. The new patients of 2010 and 2011 were examined. In particular, the “primary with progression” group includes:Patients who have undergone chemotherapy or radiotherapy within 14 months and over 18 months after surgery;Patients with secondary BC in the follow-up period;Patients with at least 10 accesses in DH for chemotherapy (DRG 410).

Alternatively, the “secondary BC” group includes patients with a secondary BC hospitalization and those with no surgery within 365 days from the first hospitalization.

### 2.2. Estimation of Social Security Costs

In Italy, the Social Security System (SSS) is characterized by a dual structure that includes, on one hand, welfare and civil incapacity care benefits, and on the other, social security benefits in a narrow sense. The latter were taken into consideration in this study. With regard to social security benefits in a narrow sense, the SSS offers economic benefits for workers with disabilities and suffering from chronic physical and/or mental incapacity, largely financed by their contributions. Specifically, all work categories registered with the National Institute of Social Security are entitled, in case of an accident or illness, to qualify, following an application, for one of the two social security benefits provided: the Disability Benefit (DB), for those whose work capacity is reduced to less than a third (disability between 67% and 99%); and the Incapacity Pension (IP) in favor of those for whom the absolute and permanent impossibility to carry out any work activity (100% disability) is ascertained (Considering the EU definition of the benefits analyzed, DB corresponds to the definition of Ordinary Incapacity Benefits of the European Commission, while IP corresponds to the Disability Pension [14]). Law no. 222/84 [15] sets the requirements for access to the social security benefits being analyzed. To estimate the social security costs related to breast cancer, information from the database of the National Social Security Institute was used, containing the number of requests submitted each year from 2006 to 2015, to determine each benefit and the related judgments (approval or rejection) expressed by medical managers. These include the indication of the prevailing diagnosis and any secondary diagnosis, based on the international classification of diseases, ninth revision (ICD-9-CM). Following an overall assessment of the physical and mental health of the applicant, the Medical–Legal Centres of the National Social Security Institute approve the request, providing the benefit based on the presence of one or more disabling diseases. The assessment is based exclusively on medical forensic criteria and does not include any examination of socio-economic or other types of factors.

As for the costs of hospitalizations, the diagnosis codes ICD-9-CM 174.xx, 233.0, and 198.81 were used to select the approved requests of the two types of social security benefits, for workers with a primary or secondary diagnosis of primary, in situ, or secondary carcinoma.

On the basis of the approved requests, the beneficiaries of the diseases in question were first estimated (i.e., the number of DBs and IPs actually provided by the National Institute for Social Security each year) and then the costs were estimated. Following the methods of Russo et al. (2015), social security beneficiaries and costs were estimated through a probabilistic model with a Monte Carlo simulation [16].

The estimate of the costs per type of benefit was conducted on the basis of the average monthly values per kind of scheme (Table 1).

## 3. Results

### 3.1. Hospitalization Costs

The results will be presented separately from the point of view of the NHS and that of the social security system. As far as the prevalence rate is concerned (Figure 1), on average, 24.3 per 10,000 resident women are found to have a primary/in situ carcinoma, slightly decreasing during the study period, and 2.4 per 100,000 resident women have a secondary carcinoma. Similarly, in this case, we observe a decreasing trend (2.7 per 100,000 in 2008 to 1.9 per 100,000 in 2016).

These prevalence rates translate into more than 544,000 subjects who, from 2008 to 2016, were hospitalized at least once for breast cancer—specifically, for one of the three types of cancer or for non-surgical treatments, such as chemotherapy or radiotherapy. The annual number of these subjects, as shown in Figure 2, is steady over the reference period. In particular, the number of women hospitalized for breast cancer was, on average, over 75,000 per year, of whom 99.2% were diagnosed with primary or in situ cancer, while the remaining 0.8% were diagnosed with secondary cancer.

Furthermore, the analysis of HDRs showed a decreasing trend in the number of hospital admissions (Figure 2). The annual value of hospitalizations, amounting to more than one million from 2008 to 2016, dropped, despite the number of subjects being steady over time. In particular, hospitalizations amounting, on average, to almost 117,000 every year, decreased by 22% from about 132,000 in 2008 to 103,500 in 2016. On average, each patient underwent 1.9 hospitalizations.

From 2008 to 2016, the annual expense of the direct costs attributable to these hospitalizations dropped by more than 17%, in line with their negative trend. During the reference period, the annual average cost amounted to more than €280 million, thus corresponding to a total expense of over €2.5 billion (Figure 3).

Finally, the analysis over the five-year follow-up period showed significant differences between the three groups of patients analyzed. In particular, we detected an average cost per patient of €3322 for patients with primary BC with no progression, a cost of €10,111 for patients with primary BC with progression, and a cost of €5883 for secondary BC.

### 3.2. Beneficiaries and Social Security Costs

Considering the results related to the provision of social security benefits by the National Social Security Institute, Figure 4 shows the trend of beneficiaries over time (i.e., the benefits provided each year and the related costs for primary, in situ, and secondary carcinoma). The analysis shows annual means of about 29,000 beneficiaries of social security benefits, of which 90% received DB. Overall, the annual number also increased by 13% between 2009 and 2015 (+14% for DBs and +8% for IPs).

These benefits caused an average annual expense of over €257 million, of which 85% is attributable to the DBs and 15% is attributable to the IPs. Moreover, in line with the increase in demand, the economic weight of both services increased by 32% and 19%, respectively, over the period considered. On the other hand, considering the different percentage distributions of the benefits by type of cancer—calculated on average over the reference period—the incapacity pensions have the lowest number of beneficiaries, as they identify those individuals who, due to the diagnosis of breast cancer, have a complete inability to work. However, while these beneficiaries do not exceed, on average, from 2009 to 2015, 10% of the patients with primary cancer, the percentage rises above 30% in women diagnosed with secondary cancer. This results in an average annual social security cost per patient with primary cancer of €8828, which rises to €9780 for patients with secondary cancer. This means that from the perspective of the Social Security System, the cost of a patient is almost as high as for progressed cancers, even though she is less expensive to treat. This occurs because there is a relatively small difference between the amounts of the two economic benefits provided by the National Social Security Institute.

Finally, analyzing the overall economic impact of breast cancer from the point of view of the NHS and the social security system (Figure 5), the average annual total cost reached almost €539 million. As previously noted, all these costs are due to primary/in situ cancer which involves an annual cost of €278 million, €220 million, and €37 million, respectively, for hospitalizations, DBs, and IPs. For the same cost items, secondary cancer has an annual weight of €2.7 million, €420,000, and about €265,000, respectively.

Figure 6 details the percentage distribution of these costs among the types of tumors analyzed and the cost categories considered. This analysis shows that the direct costs, particularly the costs linked to hospitalizations, are those with the most weight (52%). The two social security benefits cover 48% of the total costs, of which 41% is attributable to the disability benefits and 7% to the incapacity pensions.

## 4. Discussion

The objective of the study was to estimate the direct costs incurred by the NHS for mammary carcinoma related to hospitalizations and the cost of the National Social Security Institute for the provision of social security benefits, such as DBs and IPs. The main objective of the study was to provide a measure in terms of the economic impact of one of the most widespread diseases in the world, especially among women, in line with the group of diseases to which it belongs. Therefore, its economic impact is very significant. The cost items analyzed in the study have been investigated considering that both direct and social security costs weigh on citizens. Specifically, the former are funded with taxes, while the latter is funded by contributions paid by the workers who joined the National Social Security Institute pension scheme.

The analysis also aimed to highlight how the differences between primary/in situ BC and secondary BC (both ex novo and recurrence), from the point of view of the progress of the disease, also produce their economic effects, both in terms of treatment and benefits related to the degree of disability. The analysis of BC patients for a five-year follow-up period showed that primary BC results in a lower cost (€3322) than both progressed (€10,111) and secondary BC (€5883).

Hospitalizations, amounting on average to 117,000 each year, from 2008 to 2016 dropped by 22%, although the number of subjects (about 75,000 each year) is steady over the period considered. A total of 99.2% of the subjects were diagnosed with primary/in situ BC, while only 0.2 were diagnosed with secondary BC. Just like hospitalizations, the annual expense also recorded a 17% contraction. In the reference period, the annual average cost amounted to more than €280 million, corresponding to a total expense of over €2.5 billion.

On the other hand, the number of beneficiaries of social security benefits (29,000 each year, total) is increasing for both DBs (+14%) and IPs (+8%). On average, over the period considered, the benefits paid are 90% for DBs and 10% for IPs. This result is due to the prevalence of cases with a diagnosis of primary/in situ BC, theoretically corresponding to a certain degree of partial disability and thus to a disability benefit. On the other hand, with reference to secondary breast cancer, IPs have a greater weight on all the benefits, as they are provided with 100% incapacity and at a more advanced stage of the disease.

Moreover, in line with the increase in demands, the economic weight of both benefits increased by 32% and 19%, respectively, over the period considered. In total, the average annual expense is over €257 million, of which 85% is attributable to DBs and 15% is attributable to IPs. The total economic impact is €539 million each year, of which 52% corresponds to hospitalizations, 41% to DBs, and 7% to IPs.

Our study confirms that breast cancer costs increase as the stage of the disease progresses, as also reported in some previously mentioned studies. In fact, when considering secondary breast cancer, this is associated with a lower total cost given the smaller number of patients, but with a higher cost per patient due to the greater severity of the disease. We have observed this relationship between costs and the stage of disease in both hospitalization costs and social security costs.

The analysis shows some limitations. With reference to the impact of breast cancer in terms of direct healthcare costs, in our analysis, we considered only the costs related to hospitalizations (ordinary hospitalizations and day-hospital admissions) due to the availability of the national HDR database, which refers to hospital care. On the contrary, it was not possible to include the pharmaceutical and outpatient healthcare costs, as there is no national database in Italy comparable to that for hospital care. Similarly, the DRG tariff represents a mean value of hospital expenditure for all hospitalization with the same DRG, but it was impossible to have a breakdown by the individual cost items which are included (surgeries, drugs administration, hospital staff, and materials). However, hospital care costs account for a very large proportion of the direct health care costs associated with breast cancer treatment [17,18].

Therefore, we believe that, by estimating the costs of hospitalizations and social security costs, the study provides a frame and a measure of the burden of breast cancer in Italy, taking into account two important perspectives from which some important characteristics of the disease itself emerge. The study can also be considered innovative in terms of results achieved and data provided and may represent a starting point for future developments to broaden the perspectives of analysis and its time horizon. To our knowledge, this is the first time that direct healthcare costs for hospitalizations and social security costs are jointly considered. A link and a comparison between the two considered cost items may be an interesting, as well as innovative, approach.

Moreover, results show that social security costs, often not taken into consideration in this type of analysis, involve a not negligible economic burden and, in addition to direct healthcare costs, it shall be considered by decision-makers in health planning.

## 5. Conclusions

The results confirm that today, due to both screening programs allowing an early diagnosis and early care of the patients, BC mortality is undergoing a contraction. The high probability of survival, even 10 years after the diagnosis, allows patients to request and access the social security benefits. Considering today’s context, in which BC represents a less dangerous threat than other diseases of the same group, this phenomenon can justify, on the one hand, the decreasing trend in the annual number of admissions, and on the other, the high number of DBs and the higher growth recorded for this social security benefit compared to IPs [19,20,21,22,23,24,25,26].

We believe that the analysis has achieved the objective of providing a framework for quantifying the costs of BC from different perspectives, concerning both the treatment of the disease and the economic benefits related to the years of post-diagnosis. The study can also be a starting point for further discussion. In fact, only the social security costs are taken into consideration, leaving aside those of a welfare nature (i.e., those related to civil invalidity).

## Figures and Tables

**Figure 1 ijerph-18-09005-f001:**
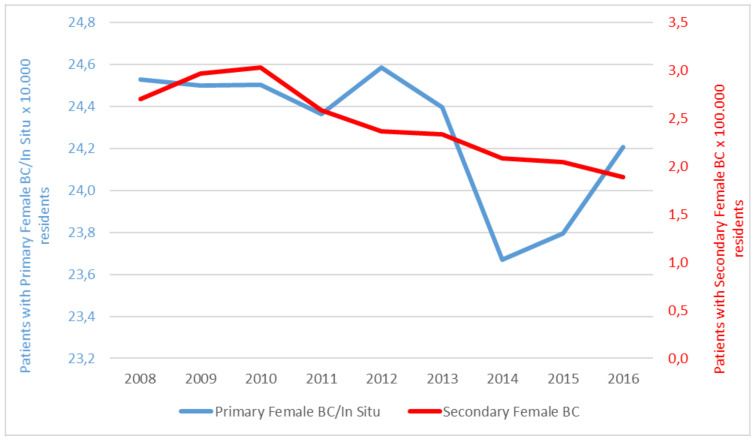
Primary or secondary breast cancer prevalence rate—Italy, 2008–2016.

**Figure 2 ijerph-18-09005-f002:**
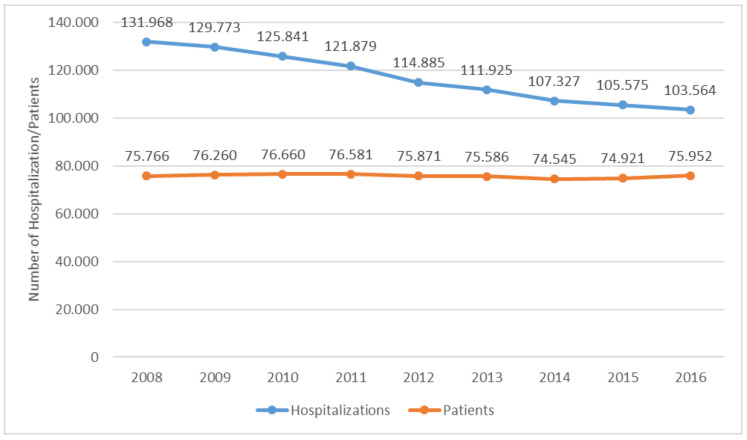
Number of patients and hospitalizations due to breast cancer per year—Italy, 2008–2016.

**Figure 3 ijerph-18-09005-f003:**
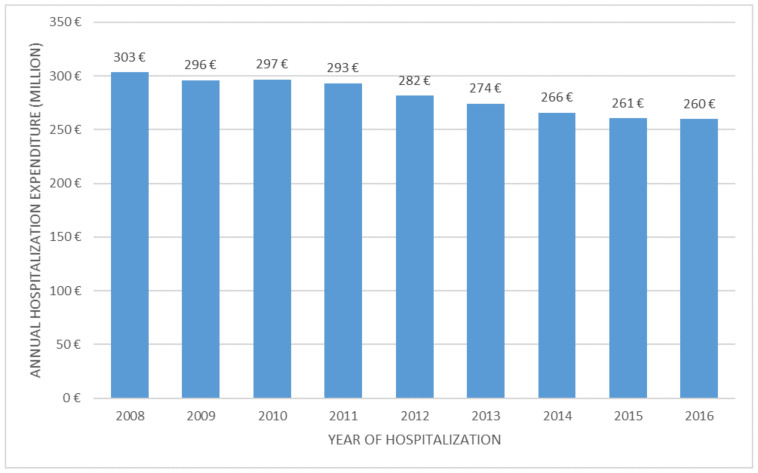
NHS average annual expense related to patients with hospital admissions for breast cancer—Italy, 2008–2016.

**Figure 4 ijerph-18-09005-f004:**
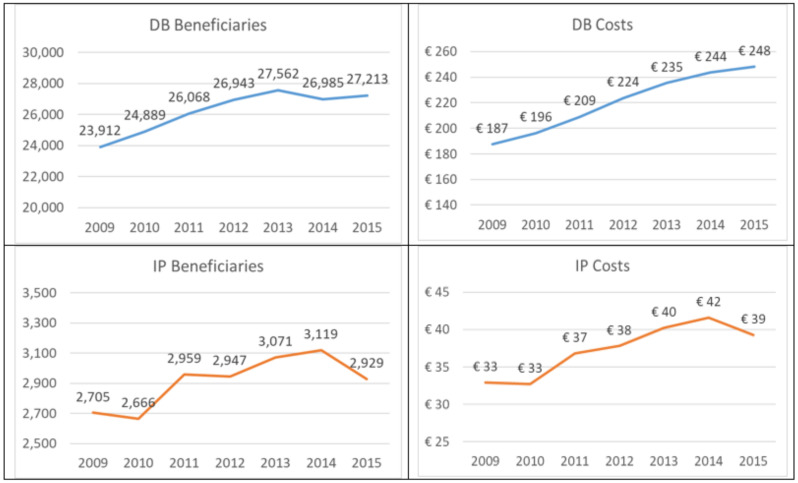
Trends of beneficiaries and costs (values in millions) for DBs and IPs for primary, in situ, and secondary cancer.

**Figure 5 ijerph-18-09005-f005:**
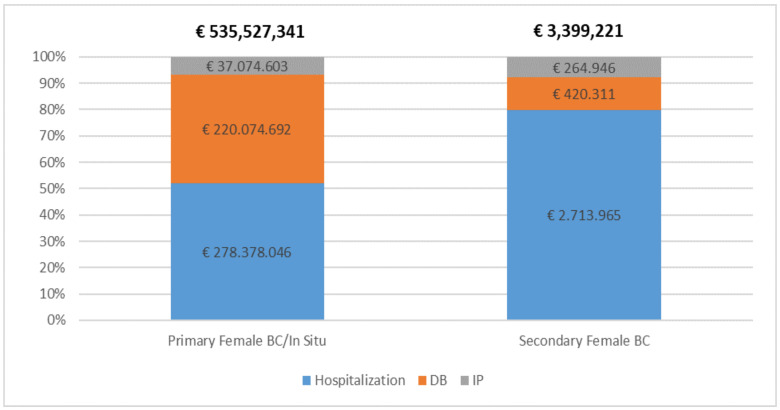
Distribution of the average annual cost of hospitalizations and social security benefits for primary/in situ and secondary cancer.

**Figure 6 ijerph-18-09005-f006:**
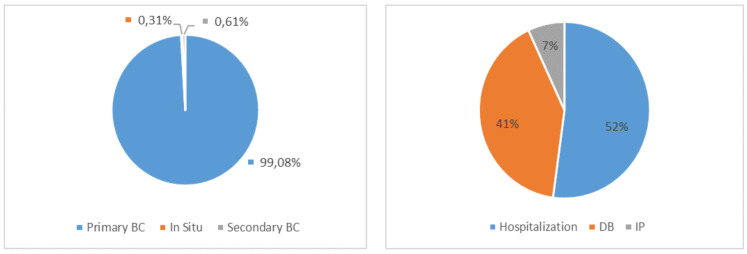
Percentage distribution of annual cost of breast cancer by type of disease and cost category.

**Table 1 ijerph-18-09005-t001:** Disability benefits and Incapacity Pensions: total benefits provided and average monthly amounts by type of benefit, 2009–2015. National Social Security Institute Statistical Observatory.

Type of Subject	2009				2010			
	Disability Benefit		Incapacity Pension		Disability Benefit		Incapacity Pension	
	Benefits Provided	Average monthly Value	Benefits Provided	Average monthly Value	Benefits Provided	Average monthly Value	Benefits Provided	Average monthly Value
Pension scheme for Employees (including separate pension schemes)	250,782	€632.0	58,388	€1018.0	254,88	€634.2	59,579	€1024.0
Self-employed pensions	115,183	€541.9	22,600	€728.1	114,781	€549.6	22,919	€739.0
Replacement pension schemes	5	€1192.9	10	€2610.5	10	€890.3	14	€2773.7
Supplementary pension schemes	-	-	-	-	-	-	-	-
Separate pension scheme for “para- subordinate” workers	758	€146.2	140	€450.4	913	€163.5	156	€463.8
Other schemes and optional insurance	-	-	-	-	-	-	-	-
Total/Weighted average	366,728	€602.7	81,138	€936.5	370,492	€606.8	82,668	€944.2
Type of subject	2011				2012			
	Disability Benefit		Incapacity Pension		Disability Benefit		Incapacity Pension	
	Benefits Provided	Average monthly Value	Benefits Provided	Average monthly Value	Benefits Provided	Average monthly Value	Benefits Provided	Average monthly Value
Pension scheme for Employees (including separate pension schemes)	259,009	€642.0	60,560	€1036.2	263,482	€665.0	61,042	€1055.6
Self-employed pensions	114,801	€562.8	23,118	€752.1	113,549	€581.3	22,908	€773.9
Replacement pension schemes	9	€694.8	12	€2894.7	12	€869.8	828	€1924.6
Supplementary pension schemes	-	-	-	-	-	-	-	-
Separate pension scheme for “para- subordinate” workers	1031	€183.8	161	€480.8	1116	€208.6	168	€514.9
Other schemes and optional insurance	-	-	-	-	-	-	-	-
Total/Weighted average	374,850	€616.5	83,851	€957.0	378,159	€638.5	84,946	€987.1
Type of subject	2013				2014			
	Disability Benefit		Incapacity Pension		Disability Benefit		Incapacity Pension	
	Benefits Provided	Average monthly Value	Benefits Provided	Average monthly Value	Benefits Provided	Average monthly Value	Benefits Provided	Average monthly Value
Pension scheme for Employees (including separate pension schemes)	268,147	€682.3	60,963	€1076.1	267,419	€719.8	61,000	€1093.1
Self-employed pensions	114,901	€602.9	22,783	€794.1	112,850	€639.7	22,518	€810.1
Replacement pension schemes	9	€1004.5	879	€1933.8	8	€882.0	928	€1950.7
Supplementary pension schemes	-	-	-	-	-	-	-	-
Separate pension scheme for “para- subordinate” workers	1249	€227.7	189	€512.8	1291	€265.4	211	€545.7
Other schemes and optional insurance	-	-	-	-	-	-	-	-
Total/Weighted average	384,306	€657.1	84,814	€1008.0	381,568	€694.5	84,657	€1025.8
2015								
Type of subject	Disability Benefit		Incapacity Pension					
	Benefits Provided	Average monthly Value	Benefits Provided	Average monthly Value				
Pension scheme for Employees (including separate pension schemes)	271.184	€725.4	61.305	€1098.8				
Self-employed pensions	112.835	€648.5	22.458	€812.6				
Replacement pension schemes	8	€883.2	958	€1965.9				
Supplementary pension schemes	-	-	.	.				
Separate pension scheme for “para- subordinate” workers	1.393	€283.8	216	€562.7				
Other schemes and optional insurance	-	-	.	.				
Total/Weighted average	385.420	€701.3	84.937	€1031.5				

## Data Availability

Data available on request due to restrictions. The data presented in this study are available on request from the corresponding author.

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
