# Peer review of "An Analysis of the Social and Economic Costs of Breast Cancer in Italy"

_ijerph, 2021, doi:10.3390/ijerph18179005_

Round 1

Reviewer 1 Report

The authors aimed to investigate the socio-economic burden of breast cancer in Italy from the National Health Service (NHS) and the government perspective (costs borne by the social security system). They analyzed the costs incurred by the NHS from 2008 to 2016 and by the National Social Security Institute (INPS) from 2009 to 2015 (costs of social security benefits) for patients with breast cancer. They found that an advanced stages of breast cancer was associated with a higher cost.

Here are my comments:

-The conclusion of this paper does not seem novel. It is a fact that advanced stages of illnesses are costly.

-The article is poorly written. For example, the introduction section starts with

The objective of the study was to estimate the direct costs incurred for breast cancer related to hospitalizations and social security benefits, considering both the NHS and ….

-There is not clear how healthcare system works in Italy.

-The authors have some data on costs of breast cancer in Italy but is poorly written and presented. 

Author Response

Thank you for your observation, for sure the treatment of advanced stages of illnesses is more expensive. We think that the contribution of our study is to give a frame about breast cancer considering the progression of the disease and different perspective. Even though advanced stages of illnesses are characterized by a greater cost per patient, primary BC, which is more widespread, involves a higher total expenditure from both the analysed perspectives.

Moreover, to our knowledge this is first time social security costs have been included in this kind of analysis. We believe that a comparison between direct costs linked to hospitalizations (often considered in other studies) and social security costs may be an interesting and innovative approach. Again, in Italy social security expenditure is a component of public expenditure and it was our intention to provide a general measure of this cost item. Therefore, it is part of the government perspective as it shall be considered by decision-makers in health planning.

Finally, considering social security benefits, we think that it is very important the opposite trends of DBs and IPs in terms of beneficiaries and costs. The growth of DBs and the reduction of IPs, confirm that breast cancer is becoming less disabling over time. Submitting a claim for a DB means being able to work, even if with a reduced work ability and productivity.

The following sentence has been added in the discussion section:

“To our knowledge this is the first time that direct healthcare costs for hospitalizations and social security costs are jointly considered. A link and a comparison between the two considered cost items may be an interesting, as well as innovative, approach. Moreover, results show that social security costs, often not taken into consideration in this type of analysis, involve a not negligible economic burden and, in addition to direct healthcare costs, it shall be considered by decision-makers in health planning.”

Rew 1

The authors aimed to investigate the socio-economic burden of breast cancer in Italy from the National Health Service (NHS) and the government perspective (costs borne by the social security system). They analyzed the costs incurred by the NHS from 2008 to 2016 and by the National Social Security Institute (INPS) from 2009 to 2015 (costs of social security benefits) for patients with breast cancer. They found that an advanced stages of breast cancer was associated with a higher cost.

Here are my comments:

-The conclusion of this paper does not seem novel. It is a fact that advanced stages of illnesses are costly.

Thank you for your observation, for sure the treatment of advanced stages of illnesses is more expensive. We think that the contribution of our study is to give a frame about breast cancer considering the progression of the disease and different perspective. Even though advanced stages of illnesses are characterized by a greater cost per patient, primary BC, which is more widespread, involves a higher total expenditure from both the analysed perspectives.

Moreover, to our knowledge this is first time social security costs have been included in this kind of analysis. We believe that a comparison between direct costs linked to hospitalizations (often considered in other studies) and social security costs may be an interesting and innovative approach. Again, in Italy social security expenditure is a component of public expenditure and it was our intention to provide a general measure of this cost item. Therefore, it is part of the government perspective as it shall be considered by decision-makers in health planning.

Finally, considering social security benefits, we think that it is very important the opposite trends of DBs and IPs in terms of beneficiaries and costs. The growth of DBs and the reduction of IPs, confirm that breast cancer is becoming less disabling over time. Submitting a claim for a DB means being able to work, even if with a reduced work ability and productivity.

The following sentence has been added in the discussion section:

“To our knowledge this is the first time that direct healthcare costs for hospitalizations and social security costs are jointly considered. A link and a comparison between the two considered cost items may be an interesting, as well as innovative, approach. Moreover, results show that social security costs, often not taken into consideration in this type of analysis, involve a not negligible economic burden and, in addition to direct healthcare costs, it shall be considered by decision-makers in health planning.”

-The article is poorly written. For example, the introduction section starts with

Introduction section has been reorganized, and the statement reported has been modified as follow:

“The objective of this study was to estimate the socio-economic burden of breast cancer in Italy, considering both the NHS and social security system perspective. For the first one, direct costs related to hospitalizations were considered, through a real-world data analysis, based on data from Hospital Information System (HIS). Together with healthcare direct costs, social security costs are an important component and must be considered in economic evaluations to define the total burden of the disease. This analysis was based on Disability Insurance awards database and estimated the number of disability benefit receipts suffering from breast cancer, and related costs.

Then, the analysis was focused considering separately different stages of the disease: primary/in situ BCs, which concern an initial stage of the disease, primary BC with progression, and secondary breast cancer, occurring at an advanced stage of the disease. and. Therefore, the objective was to analyse the cost difference on the basis of the presence or absence of early diagnosis and early care of the patient.”

-There is not clear how healthcare system works in Italy.

Thank you for your suggestion, in the Introduction section we added the following:

“Italian health care system is a mixed public-private system. The National Health Service guarantees universal and free of charge coverage for all citizens and legal foreign residents. It is funded by corporate and value-added tax revenues collected by the central government and distributed to the regional governments, which are responsible for delivering care. Residents receive mostly free primary care, inpatient care, and health screenings.

Since the NHS does not allow people to opt out of the system and seek only private care, substitutive insurance does not exist, and complementary and supplementary private health insurance play a limited role in the health system. On the other hand, social security system provides, upon request, economic benefits to all workers whose working capacity is reduced or absent, due to physical or mental illness, largely financed by their contributions”.

-The authors have some data on costs of breast cancer in Italy but is poorly written and presented.

We thank for this suggestion. The “Data and methods” section has been expanded focusing on data by INPS in a deeper way to better contextualize the INPS data and the social security benefits.

“In Italy, the Social Security System (SSS) is characterized by a dual structure that includes, on one hand, welfare and civil incapacity care benefits, and, on the other, social security benefits in a narrow sense. The latter were taken into consideration in this study. With regard to social security benefits in a narrow sense, the SSS offers economic benefits for workers with disability and suffering from chronic physical and/or mental incapacity, largely financed by their contributions. Specifically, all work categories registered with the National Institute of Social Security are entitled, in case of an accident or illness, to benefit, following an application, for one of the two social security benefits provided: the Disability Benefit (DB), for those whose work capacity is reduced to less than a third (disability between 67% and 99%), and the Incapacity Pension (IP) in favor of those for whom it is ascertained the absolute and permanent impossibility to carry out any work activity (100% disability) . Law no. 222/84 [13] sets the requirements for access to the social security benefits being analysed.” …… “The assessment is based exclusively on medical forensic criteria and does not include any ex-amination of socio-economic or other types of factors”.

Reviewer 2 Report

It seems to me that this study is appropriately designed and executed. I believe that this paper can be published in this journal.

However, I hope that the following point will be added to the manuscript during revision, if possible.

Discussion (line 321 - 326): "The analysis shows some limitations... ...as there is no national database in Italy similar to that for hospital care."

This description confuses me. In the method section, it is necessary to explain clearly and in detail which specific sub-type of costs are included in 'the direct cost related to hospital'.

Discussion (line 326 - 328): "However, hospital care costs... ...with breast cancer treatment."

Please add reference to support this statement.

Author Response

We have considered the costs of hospitalizations because we have the possibility of querying the national HDR (hospital discharge records) database. These data are difficult to find and very important in terms of health planning. The national HDR database refers to hospital care and contains information on all admissions to public and private hospitals throughout the country. As already written in the method section, the DRG system quantifies the hospitalization cost, including drugs, materials, hospital staff and surgeries. The DRG tariff, which is reimbursed to hospitals, represents a mean value of hospital expenditure for all hospitalization with the same DRG, but it was impossible to have a breakdown by cost item.

In Italy there is no national database for pharmaceutical care and outpatient care, but only regional databases. Therefore, it was not possible to include these cost items – net of any innovative drugs that were on the market until the end of the period covered by our study – which represent a minority share of total costs compared with hospitalizations. The lack of pharmaceutical and outpatient costs was already written in the “Discussion” section.

We have made clearer this point about direct cost related to hospitalization adding the following sentence among limitations in the discussion section:

“Also, the DRG tariff represents a mean value of hospital expenditure for all hospitalization with the same DRG, but it was impossible to have a breakdown by cost item which are included (surgeries, drugs administration, hospital staff and materials)”.

Discussion (line 326 - 328): "However, hospital care costs... ...with breast cancer treatment."

Please add reference to support this statement.

Thank you for your observation. We have added the following two references:

  • Radice, D.; Redaelli, A. Breast Cancer Management. PharmacoEconomics volume 21, pages383–396 (2003).
  • Barron, J.J.; Quimbo, R.; Nikam, P.T.; Amonkar, M.M. Assessing the economic burden of breast cancer in a US managed care population. Breast Cancer Research and Treatment volume 109, pages367–377 (2008)

Reviewer 3 Report

This paper presents data on an economic analysis of breast cancer expenses in a single country   comments below are meant to improve the quality of reporting 

The term hospital expenditures was used to describe one portion of the costs   can the  why this was selected and what expeditures are not hospitalizations?   Does this include outpatient visits?

Can the country be added to tthe title?

The line 270 includes the word weighing  Is this what is meant?

Overall the analyses are nicely done and presented.   It is not clear what unique or novel is contributed above the existing literature.   Can authors make this explicit?

Author Response

As written above, our study quantifies hospitalization costs through the DRG system. The DRG tariff, which is reimbursed to hospitals, represents a mean value of hospital expenditure for all hospitalization with the same DRG, but it was impossible to have a breakdown by cost items that are included in the tariff (drugs, surgeries, hospital staff and materials). Among the study limitations (discussion section), we reported that it was impossible to include pharmaceutical and outpatient cost due to their regional-level data administration in Italy. However, considering breast cancer, hospital care costs should account for larger proportion of the direct health care costs.

We have made clearer this point about direct cost related to hospitalization adding the following sentence among limitations in the discussion section:

“Also, the DRG tariff represents a mean value of hospital expenditure for all hospitalization with the same DRG, but it was impossible to have a breakdown by cost item which are included (surgeries, drugs administration, hospital staff and materials)”.

Can the country be added to the title?

We agree with you. The title has been changed in “An national analysis of the social and economic costs of breast cancer in Italy.”

The line 270 includes the word weighing  Is this what is meant?

Thank you for the suggestion. The sentence has been corrected as follow:

“…with the most weight”

Overall the analyses are nicely done and presented.   It is not clear what unique or novel is contributed above the existing literature.   Can authors make this explicit?

Thank you for this observation. To our knowledge this is the first time that social security data have been analysed to estimate the impact of breast cancer in terms of social security expenditure in Italy. We have included social security costs because they represent a cost item which is rarely considered in this type of analysis. Furthermore, we believe that a comparison between direct costs linked to hospitalizations (often considered in other studies) and social security costs may be an interesting and innovative approach. In Italy social security expenditure is a component of public expenditure. Therefore, it is part of the government perspective as it shall be considered by decision-makers in health planning. Moreover, both hospitalization and social security costs are based on reliable and robust national data.

The discussion section has been expanded adding the following sentence:

“To our knowledge this is the first time that direct healthcare costs for hospitalizations and social security costs are jointly considered. A link and a comparison between the two considered cost items may be an interesting, as well as innovative, approach. Moreover, results show that social security costs, often not taken into consideration in this type of analysis, involve a not negligible economic burden and, in addition to direct healthcare costs, it shall be considered by decision-makers in health planning.”

Reviewer 4 Report

Dear Authors,
An interesting and important issue, timely done. The cost and the social benefits are important and challenging and your paper is well written. However, the manuscript suffers from minor flaws that need to be addressed before it can be considered for publication.

  • For examples, how many hospitals are considered in your analysis?
  • Are there differences for different regions? could you explain better the source of data?

Author Response

Thank you for the consideration you raised. As already reported in the method section, the national HDR (hospital discharge records) database refers to hospital care and contains information on admissions to all public and private hospitals throughout the country.

For sure there are a lot of differences between Italian regions in terms of number of hospitals, both public and private structures, and health services. We think that focusing on this point could be a very interesting goal for further and future analysis, which would be possible through the data we have access to.

Some additions have been made to method and discussion section to clarify data sources and which cost are included and which are not.